# ProtoVAE: Using Prototypical Networks for Unsupervised Disentanglement

## Abstract

Generative modeling and self-supervised learning have in recent years made great strides towards learning from data in a completely *unsupervised* way. There is still however an open area of investigation into guiding a neural network to encode the data into representations that are interpretable or explainable. The problem of unsupervised *disentanglement* is of particular importance as it proposes to discover the different latent factors of variation or semantic concepts from the data alone, without labeled examples, and encode them into structurally disjoint latent representations. Without additional constraints or inductive biases placed in the network, a generative model may learn the data distribution and encode the factors, but not necessarily in a disentangled way. Here, we introduce a novel deep generative VAE-based model, ProtoVAE, that leverages a deep metric learning Prototypical network trained using self-supervision to impose these constraints. The prototypical network constrains the mapping of the representation space to data space to ensure that controlled changes in the representation space are mapped to changes in the factors of variations in the data space. Our model is completely unsupervised and requires no *a priori* knowledge of the dataset, including the number of factors. We evaluate our proposed model on the benchmark dSprites, 3DShapes, and MPI3D disentanglement datasets, showing state of the art results against previous methods via qualitative traversals in the latent space, as well as quantitative disentanglement metrics. We further qualitatively demonstrate the effectiveness of our model on the real-world CelebA dataset.

## 1 Introduction

One theory of the success of deep learning models for supervised learning revolves around their ability to learn mappings from the input space to a lower dimensional abstract representation space which are best predictive of the corresponding labels Tishby & Zaslavsky (2015). However, for the models to be robust to noise and adversarial examples, be transferable to different domains and distributions and interpretable, we need to impose additional constraints on the learning paradigm. As a promising solution to this, the models can be encouraged to focus on *all* the latent "distinctive properties" of the data distribution and encode them into a representation for downstream supervised tasks. These latent distinctive properties or *factors of variations* are the interpretable abstract concepts that describe the data.

The intuitive notion of *disentanglement*, first proposed in Bengio (2013), proposes to discover all the different factors of variations from the data, and encode each factor in a separate subspace or dimension of the learned latent representation. These disentangled representations are not only interpretable and give valuable insights into the data distribution but are also more robust for multiple downstream tasks Bengio (2013); Schoelkopf et al. (2012) which might depend only on a subset of factors Suter et al. (2019).

The problem of learning these disentangled representations in a completely *unsupervised* way is particularly challenging as we do not have access to the ground truth labels of factors nor are privy to the true number of factors or their nature. Recent works have proposed to solve this problem by training generative networks to effectively model the data distribution and in turn the factors of variations. From this generative perspective of disentanglement, higher dimensional data is assumed to be a non-linear mapping of these factors of variation, where each factor assumes different values

to generate specific examples in the data distribution. Locatello et al. (2019) intuitively characterizes representations which encode the factors as *disentangled* if a change in a single underlying factor of variation in the data produces a change in a single factor of the learned representation (or a change in the subspace of the representation that encodes that factor). Conversely, from the generative perspective, for a representation to be disentangled, a change in a single subspace of the learned representation, when mapped to the data space, must produce a change in a single factor of variation.

For this generative mapping between changes in the representation space to the changes in the factors of variations (in the data space) to be injective, we propose constrains on the changes in the factors of variations for pre-determined changes in the representation space. Each separate subspace of the representation, when changed, must map to a change in a *unique* factor of variation which in turn encourages information about the different factors to be encoded in separate subspaces of the representation. Moreover, each separate subspace must *consistently* map to a change in a single factor throughout the subspace range. This encourages the different subspaces of the representation to encode information only about a single factor of variation. The recent work of Horan et al. (2021) also demonstrated empirically that the concept of *local isometry* was a good inductive bias for unsupervised disentanglement, and it can aid generative models in discovering a "natural" decomposition of data into factors of variation. This local isometry constraint on the mapping enforces the changes in the data space to be proportional to any changes made in the representation space. In order to effectively impose the above constraints in an unsupervised manner, we turn towards deep metric learning.

In recent years, metric learning has emerged as a powerful unsupervised learning paradigm for deep neural networks, in conjunction with self-supervised data augmentation. One of the more successful metric learning models, Prototypical Networks, projects the data into a new metric space where examples from the same class cluster around a prototype representation of the class and away from the prototypes of other classes. We use this ability of the network to cluster the different changes in the data space mapped by the corresponding changes in the representation space and thereby enforce the above described constraints.

We develop a novel deep generative model, ProtoVAE, consisting of a Prototypical Network and Variational Autoencoder network (VAE). The VAE acts as the generative component, while the Prototypical Network guides the VAE in separating out the representation space by imposing the constraints for disentanglement. To learn these representations in an unsupervised way, as the prototypical network needs labeled data for clustering, we train the prototypical network using generated self-supervised datasets. To produce the self supervised dataset, we perform *interventions* in the representation space, which change individual elements of the latent space and map the intervened representations to the data space. Owing to the self-supervised training, our model is able to disentangle without any explicit prior knowledge of the data, *including* the number of desired factors.

In this work, our core contributions are:

- We design a self-supervised data generation mechanism using a VAE that creates new samples via a process of intervention to train a metric-learning prototypical network.
- We design and implement a novel model, ProtoVAE, which combines a VAE and prototypical network to perform disentanglement without any prior knowledge of the underlying data.
- We empirically evaluate ProtoVAE on standard benchmark DSprites, 3DShapes, MPI3D, and CelebA datasets, showing state of the art results.

## 2 PROTOVAE

Our proposed model consists of a VAE Kingma & Welling (2014); Rezende et al. (2014) as the base generative model (Section 2.1). The VAE consists of an inference network which encodes the data into lower dimensional latent representations and a generator network that maps the representations back into the data space. To implicitly encourage the inference network to encode disentangled representations, we impose constraints on the generative mapping from changes in the representation space to changes in the factors of variations in the data space. This generative mapping is determined by both the generator and the inference networks. To generate self-supervised data for the prototypical network, we perform interventions (Sec 2.2) which changes individual dimensions of

the representation. Given a batch of latent representations encoded by the inference network, we first intervene on a dimension of the representation by changing its value to the value of another representation from the batch for the same dimension. The original representations and the intervened representations are then mapped into the data space by the generator network and concatenated to form a pair of original data and generated data from interventions. Given that the original and the intervened representations differ in a single dimension, the generative mapping should be constrained to ensure that the corresponding pair of original and generated data differs only in a single factor of variation.

This constraint is enforced using a Prototypical network (Appendix A.2) which based on the idea that there exist an embedding in which examples from the same class cluster around a prototype representation for that class. Our proposed prototypical network (Section 2.3) takes as input pairs of data generated by the self-supervised process described above, and maps these pairs of data into a metric space in which pairs generated by intervening on the same dimension cluster together. These clusters which are identifiable with intervening dimensions in-turn become identifiable with the factors of variation that differ in value between the pair when a dimension is intervened upon. We further augment the prototypical network with a separate output head that enforces local isometry, by predicting the difference in the value of the intervened dimensions from the pair in the data space. Fig 1 gives the diagram overview of the complete model.

Lastly, for the intervened representations to be mapped into the data space such that only a *factor of variation* is changed, we constrain the generated data to lie in the true data distribution. This constraint can be effectively enforced in the representation space by minimizing the distance between the distribution of the original representations of the inference network and the intervened representations such that the generator network maps both the distributions to the true data distribution. We do so by training a discriminator network (Section 2.4) in the representation space to distinguish between the original and the intervened representations. The inference network which generates the original representations is then trained to fool the discriminator thus effectively bridging the distance between the distributions.

## 2.1 VARIATIONAL AUTOENCODER

The base generative model ~~which~~ consists of an inference network $q_\phi : \mathbb{R}^D \to \mathbb{R}^d$ that encodes the data $x$ to a lower dimensional representation $z$ and a generator network $p_\theta : \mathbb{R}^d \to \mathbb{R}^D$ which reconstructs the data $\hat{x}$ from the representations $z$. The inference and the generator network are trained together to maximize the evidence lower bound (ELBO) of the data log-likelihood as in equation 1.

$$\max_{\theta,\phi} \mathcal{L}_V(\theta, \phi) = \mathbb{E}_{q_\phi(z|x)}[\log p_\theta(x|z)] - \text{KL}(q_\phi(z|x)||p(z)) \qquad (1)$$

Maximizing the first term of equation 1 ensures that the latent representation encodes all the information needed to faithfully reconstruct the data from the representation alone. This ensures that the representations encode all the different factors of variations in the data. The KL divergence term creates an information bottleneck which enforces optimal, compact encoding of the data by enforcing the posterior distribution to be similar to the independent, non-informative prior distribution. For more details please refer to Appendix A.1.

## 2.2 SELF-SUPERVISED DATA GENERATION

The prototypical network works to cluster changes in the factors of variations in the data space and provides gradients to the generator and inference network to better separate out the factors in the representation space. To do so, the prototypical network requires a set of supervised examples, called the support set, from each class to compute a prototype around which examples from the same class cluster. Furthermore, a supervised query set is required to compute the distance of query examples from the target prototypes and the subsequent loss is used to update the network. For learning disentangled representations in an *unsupervised* way, we propose to generate these support and query sets using self-supervision. We describe the full algorithm in Appendix B.1.

Given a batch of data $x$, we first use the inference network to encode the data into the representation space $z \in \mathbb{R}^d$. Following Suter et al. (2019), we define *interventions* as the act of changing the value of a single dimension of the representation $k \in_R [d]$ while keeping the values of the other

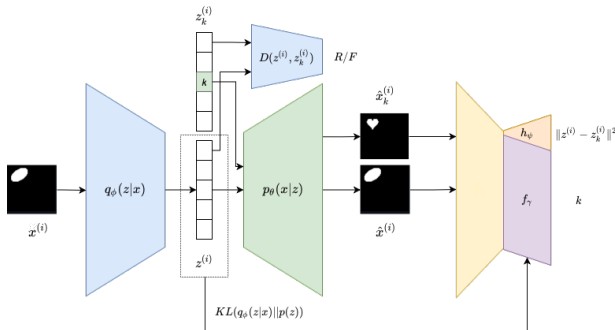

Figure 1: Architecture of our model consisting of a VAE, a discriminator and a Prototypical network. The representation $z$ from the inference network of the VAE (Sec 2.1), is changed at a particular dimension $k$ to get the intervened representation $\hat{z}_k$. A discriminator (Sec 2.4) is trained to distinguish between $z$ and $\hat{z}_k$ and the inference network is updated to fool the discriminator. $z$ and $\hat{z}_k$ are passed to the generator network to map it to the reconstructed data $\hat{x}$ and the intervened data $\hat{x}_k$. The original and the intervened data are concatenated to form the pair $(\hat{x}, \hat{x}_k)$, which is then passed to the prototypical network. The prototypical network (Sec 2.3) maps the pair closer to other pairs with the same dimension intervened. The prototypical network is updated by it's ability to correctly predict the intervened dimension of the query examples and the magnitude of the change $\|z - \hat{z}_k\|$.

dimensions the same. We change the value of the intervened dimension to another example's value for the same dimension. The result of intervening on representation $z$ in dimension $k$ produces the intervened representation $\hat{z}_k$. The representation and the intervened representation are then mapped by the generator network to $\hat{x}$ and $\hat{x}_k$ respectively. This pair of $(\hat{x}, \hat{x}_k)$ forms the input data for the prototypical network with the intervened dimension $k$ being the label and the difference in the representation space $|z - \hat{z}_k|$ as the label for the isometry head. The representation $z$ is intervened upon every dimension to generate $d$ support sets and is also intervened upon one dimension chosen uniformly from $[d]$ to generate the query set. The self-supervised data generation algorithm takes in a batch of data $x$ and outputs $d$ support sets $S = \{S_1, \cdots, S_d\}$, one query set $Q$, labels for the query set $L$ and labels for training the isometric head $I$.

## 2.3 PROTOTYPICAL NETWORK

Our proposed prototypical network maps a pair of data generated by intervening on a single dimension of the representation to a lower dimensional metric space. By mapping *a pair of data* to the metric space, the prototypical network can focus on the factor differing in value between the pair while being invariant to the values of the other non-differing factors. Critically, the factor differing in value remains the same across pairs of different examples when the same dimension is intervened upon and hence should be mapped closer in the metric space. Thus, comparing a pair of data allows the prototypical network to focus on the *change* or *difference* that was brought about by the intervened dimension, and makes the central focus of the losses this change.

The prototypical network first takes in elements of the generated support set $S$ (described in Section 2.2) and computes an $m$-dimensional representation through the embedding function $f_\gamma : \mathbb{R}^D \times \mathbb{R}^D \to \mathbb{R}^m$. In this $m$-dimensional space, the prototypical network computes a prototype embedding $c_k$ for each element in the support set $S_k \in S$ using eq. 2:

$$c_k = \frac{1}{|S_k|} \sum_{s_k^{(i)} \in S_k} f_\gamma(s_k^{(i)}) \tag{2}$$

While the support set is used to compute the prototypes, the query set is used to compute the loss by calculating the distance of its embeddings in the metric space to the target prototypes. For each dimension of the representation to encode information about a *unique* factor of variation, each dimension when intervened upon and mapped to the data space must change a different factor of

variation. Thus embeddings of pairs of data generated with the same intervening dimension of the representation must cluster closer in the metric space and away from the clusters of other dimensions. To enforce this, we introduce the *uniqueness* loss which is computed for each query $q^{(i)}$ example by calculating the negative log-likelihood of the true class $l$ as in eq. 3:

$$\min_{\gamma,\phi,\theta} \mathcal{L}_U(\gamma,\phi,\theta) = -\frac{1}{|Q|} \sum_{q^{(i)} \in Q} \log p_\gamma(t=l|q^{(i)}) \cdot \text{KL}(q_\phi(z_l|x)||p(z)) \tag{3}$$

where the probability of each class $p_\gamma(t=l|q^{(i)})$ is calculated as a distribution over the Euclidean distance $d$ to the prototypes as in eq. 4.

$$p_\gamma(t=l|q^{(i)}) = \frac{\exp\left(-d(f_\gamma(q^{(i)}), c_l)\right)}{\sum_{k'} \exp\left(-d(f_\gamma(q^{(i)}), c_{k'})\right)} \tag{4}$$

The loss for every intervening dimension of the query examples is multiplied by the KL-divergence of that dimension, averaged for the batch of examples. This ensures that the loss for the intervening dimensions is scaled by amount of information encoded by that dimension. For the dimensions that do not encode any information, and hence do not change any factor of variation upon intervention, the corresponding loss is scaled by zero. This is important as we do not need any prior assumptions on the dimension of the representation needed to encode all the factors and the VAE can find the right number of dimensions needed.

In addition to the uniqueness loss, we want each dimension to consistently encode only a single factor of variation. When the representation $z$ is first intervened on dimension $k$ and mapped to the data space it makes a certain change in factor. When the representation is intervened again at dimension $k$ by a different amount and mapped to the data space it should produce a change in the same factor, irrespective of the amount it was changed. When passed in to the prototypical network, the pair of data generated by the original $z$ and intervened $\hat{z}_k$ must be embedded in the prototypical metric space closer to the pair generated by the representation intervened in the same dimension by a different amount. To enforce this we introduce the *consistency* loss in eq. 5 where the prototypes are replaced by the embeddings of the same example in the support set.

$$\min_{\gamma,\phi,\theta} \mathcal{L}_C(\gamma,\phi,\theta) = -\frac{1}{|Q|} \sum_{q^{(i)} \in Q} \log r_\gamma(t=l|q^{(i)}) \cdot KL(q_\phi(z_l|x)||p(z)) \tag{5}$$

With $r_\gamma$ calculated as follows:

$$r_\gamma(t=l|q^{(i)}) = \frac{\exp\left(-d(f_\gamma(q^{(i)}), f_\gamma(s_l^{(i)}))\right)}{\sum_{k'} \exp\left(-d(f_\gamma(q^{(i)}), f_\gamma(s_{k'}^{(i)}))\right)} \tag{6}$$

The consistency loss and the uniqueness loss are added together to get a combined prototypical loss eq. 7

$$L_P(\gamma,\phi,\theta) = L_C + L_U \tag{7}$$

As an additional inductive bias, as proposed in Horan et al. (2021), we constrain the generative mapping between original and intervened representation $(z, \hat{z}_k)$ and the generated pair $(\hat{x}, \hat{x}_k)$ to be locally isometric Donoho & Grimes (2003). Thus the factor changed in $\hat{x}_k$ when compared with $\hat{x}$ must differ in value proportional to the corresponding change in dimension $k$ of $z$ and the intervened $\hat{z}_k$. This serves as an imperative inductive bias for unsupervised disentanglement.

The additional head $h_\psi : \mathbb{R}^D \times \mathbb{R}^D \to \mathbb{R}^d$, when given a pair of data, is trained to predict the difference in the values for the all the dimensions of $z$ and $\hat{z}_k$ through the loss function in equation 8.

$$\min_{\psi,\theta,\phi} \mathcal{L}_I(\psi,\theta,\phi) = \|h_\psi((\hat{x}, \hat{x}_k)) - |z - \hat{z}_k|\|^2 \tag{8}$$

The training data for the isometry head generated in a self supervised manner as described in section 2.2 where the support set $S$ consists of data pairs and the set $I$ consists of the corresponding targets. In the final implementation, $f_\gamma$ and $h_\psi$ share all hidden convolutional layers and differ only in the final fully connected layer.

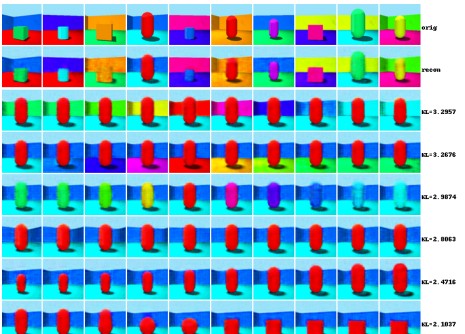 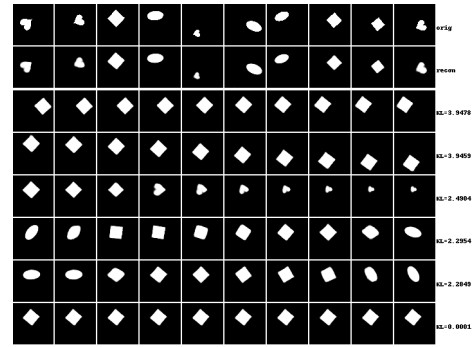

Figure 2: A comparison of latent traversals in latent space for the 3DShapes and Dsprites dataset. *Left:* 3DShapes, *Right:* Dsprites. Proto-VAE produces smooth, disentangled latent representations. Row 1 and 2 are some sample original images, and their reconstructions generated by our model, respectively. Rows 3 downward are the traversals for each latent element, as detailed below. For 3DShapes, we actually see a near-perfect traversal across all of the known factors of variation.

## 2.4 REPRESENTATION SPACE DISCRIMINATOR

The above described process does not guarantee that the intervened representations will lie in the same distribution, as the original representations which can lead to the intervened representations being mapped in the data space away from the true data distribution. To minimize the distance between the distribution of the original representations $z$ and the intervened representations $\hat{z}$, we train a separate discriminator network in the representation space to distinguish the intervened representations from the original representations by minimizing the loss in eq. 9.

$$\min_{w} \mathcal{L}_D(w) = -[\mathbb{E}_{\hat{z}}[\log(D_w(\hat{z}))] + \mathbb{E}_z[\log(1 - D_w(z))]] \tag{9}$$

Note here that the real samples for the discriminator are the intervened representations and the fake samples are the original representations. Similar to the training of the generator in a Generative Adversarial Network (GAN) Goodfellow et al. (2014), the inference network, which generates these representations, is trained to fool this discriminator by minimizing the loss in equation 10. Since each dimension in the original representation can be independently changed to obtain the intervened representation, the inference network is encouraged encode representations such that the total correlation loss $\mathrm{KL}(q(z) \| \prod_i q(z_i))$ is minimized as proposed in Kim & Mnih (2019).

$$\min_{\phi} \mathcal{L}_E(\phi) = \mathbb{E}_z[\log(1 - D_w(z))] \tag{10}$$

The final objective (equation 11) of our method is a weighted sum of the different losses, and is optimized by the network parameters of the VAE and the prototypical network corresponding to each loss.

$$\min_{\phi, \theta, \gamma, \psi} \mathcal{L} = -\mathcal{L}_V(\phi, \theta) + \alpha \mathcal{L}_E(\phi) + \lambda \mathcal{L}_P(\gamma, \phi, \theta) + \kappa \mathcal{L}_I(\psi, \phi, \theta) \tag{11}$$

## 3 EMPIRICAL EVALUATION

To empirically evaluate our method, we perform both quantitative and qualitative evaluation on two synthetic datasets and one real dataset with known factors of variation and qualitative evaluation on CelebA dataset Liu et al. (2015). The two synthetic and one real datasets are generated from independent ground truth factors of variation; DSprites Matthey et al. (2017) binary 64 x 64 images with 5 factors of variation: 3 shapes, 6 scales, 40 orientations, 32 x-positions and 32 y-positions; 3D Shapes Burgess & Kim (2018) 64 x 64 x 3 color images with 6 factors of variations: 4 shapes, 8 scales, 15 orientations, 10 floor colors, 10 wall colors and 10 object colors; MPI3D real Gondal et al. (2019) 64 x 64 x 3 color images with 7 factors of variations: 6 colors, 6 shapes, 2 sizes, 3 camera heights, 3 background colors, 40 horizontal axis, 40 vertical axis, and one real world dataset;

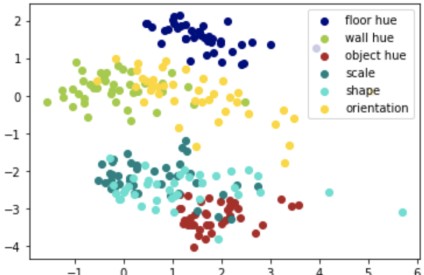 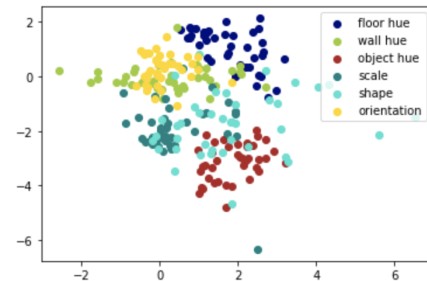

Figure 3: Output of the prototypical network with embedding dim $m = 2$ when the input is real pairs of data with a single factor with differing values amongst the pair on the *3DShapes* dataset. Each color corresponds to a different uniquely changed factor of variation. The network clusters the changes correctly on the pairs from the original dataset. This suggests that the prototypical network is clustering pairs of images based on the changed factor of variation. *Left*: $\lambda = 10$. *Right*: $\lambda = 5$.

CelebA. The details of the architecture for the different components and the corresponding hyperparameters are listed in the supplementary material (Appendix B.2) and (Appendix B.3) respectively.

| DATASETS | DSPRITES | | | 3DSHAPES | | |
|---|---|---|---|---|---|---|
| MODEL | FVAE | DCI | MIG | FVAE | DCI | MIG |
| $\beta$-VAE | $0.51 \pm .10$ | $0.23 \pm .10$ | $0.15 \pm .10$ | $0.81 \pm .10$ | $0.44 \pm .17$ | $0.28 \pm .18$ |
| ANNVAE | $0.70 \pm .10$ | $0.28 \pm .10$ | $0.23 \pm .10$ | $0.84 \pm .09$ | $0.46 \pm .16$ | $0.31 \pm .15$ |
| $\beta$-TCVAE | $0.68 \pm .10$ | $0.35 \pm .06$ | $0.17 \pm .09$ | $0.88 \pm .07$ | $0.63 \pm .10$ | $0.40 \pm .18$ |
| FVAE | $0.74 \pm .06$ | $0.38 \pm .10$ | $0.28 \pm .09$ | $0.81 \pm .06$ | $0.47 \pm .12$ | $0.33 \pm .14$ |
| GR-FVAE | $\mathbf{0.75 \pm .08}$ | $0.41 \pm .07$ | $0.31 \pm .06$ | $0.79 \pm .06$ | $0.49 \pm .06$ | $0.43 \pm .11$ |
| **PROTOVAE** | $0.70 \pm .06$ | $\mathbf{0.51 \pm .04}$ | $\mathbf{0.37 \pm .09}$ | $\mathbf{0.90 \pm .06}$ | $\mathbf{0.84 \pm .07}$ | $\mathbf{0.71 \pm .11}$ |

Table 1: Various disentanglement metrics evaluated across a number of state of the art methods for the DSprites and 3Dshapes dataset. For all metrics, **higher is better.** The results for the other models are obtained using the hyperparameter settings and experimental conditions as described in Locatello et al. (2019). The scores for all the models were averaged across ten runs with different random seeds, with standard deviation shown as $\pm$. Gr-FVAE is the GroupifyVAE variant applied to the FactorVAE, as this is the closest to our model and the results for which are taken from Yang et al. (2021). The highest values in a column are written in bold. As we see, the ProtoVAE outperforms the state of the art on a majority of the metrics. Proto-VAE hyperparameters for DSprites and 3Dshapes results shown are $\{\alpha = 10, \lambda = 10, \kappa = 10\}$ and $\{\alpha = 20, \lambda = 20, \kappa = 20\}$

We qualitatively evaluate our model by intervening on the different dimensions of the learned representations and traversing the range of values of the dimension linearly in a fixed range $[-2, 2]$. A model is better disentangled if the changes made in the data space while traversing a dimension are similar to the changes in a factor of variation in the data space.

Our model both finds the correct number of factors and encodes them separately without any specific hyper-parameter tuning. From Figure 2 we can see that our method ProtoVAE, produces disentangled traversals covering both the number of factors as well as the entirety of the range of the values for the Dsprites and the 3DShapes dataset. The latent traversals on the MPI3D dataset can be found in figure 6. Our method effectively separates the factors thus disentangling the learned latent representation without compromising on the reconstruction quality as seen from row 2. Owing to the unsupervised nature our method struggles to exactly disentangle the non-isometric discrete factor of shape in the DSprites dataset. For the 3DShapes dataset, in our traversals in 2, we achieved near perfect disentanglement, completely unsupervised. In the Appendix D, we show the performance of the ProtoVAE for a subset of the Dsprites dataset with only a few factors. We show traversals from models FactorVAE and $\beta-$VAE, along with our model, with only a few isometric factors for com-

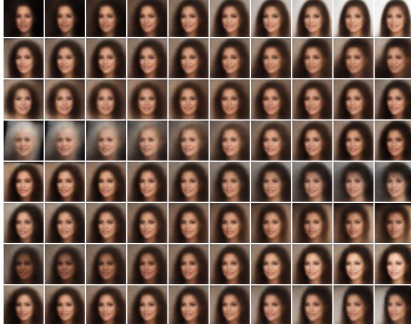

| Model | FVAE | DCI | MIG |
|-------|------|-----|-----|
| $\beta$-VAE | .41 ± .05 | .23 ± .04 | .06 ± .03 |
| AnnVAE | .29 ± .04 | .12 ± .02 | .07 ± .07 |
| $\beta$-TCVAE | .45 ± .06 | .27 ± .03 | .16 ± .03 |
| FVAE | .40 ± .04 | .30 ± .03 | **.23 ± .03** |
| DisCo | .39 ± .07 | .29 ± .02 | .07 ± .03 |
| **ProtoVAE** | **.46 ± .04** | **.38 ± .05** | **.25 ± .11** |

Figure 4: *Left:* Latent traversals on the CelebA dataset ProtoVAE successfully captures ground-truth factors of variation on real-world data. From top to bottom: background color, hairstyle, head angle, age, hairstyle, hair color, skin color, face profile. *Right:* Quantitative comparative metrics on the MPI3D dataset. ProtoVAE performs comparatively or better consistently across multiple metrics on a difficult real disentanglement dataset. See Appendix Fig. 6 for latent traversals on MPI3D. Proto-VAE hyperparameters for results shown are $\{\alpha = 10, \lambda = 2, \kappa = 2\}$. The numbers for DisCo have been borrowed from their paper Ren et al. (2021) for the VAE-based methods.

parison. Our proposed ProtoVAE is the only model that does not conflate two factors and encodes them in separate dimensions of the representation.

Furthermore, we quantitatively evaluate the learned representation by calculating state-of-the-art disentanglement metrics. We choose metrics from each of the three kinds of metrics described in Zaidi et al. (2021); Intervention-based FactorVAE Kim & Mnih (2019), Predictor-based Disentanglement-Completeness-Informativeness (DCI) Eastwood & Williams (2018) and Information-based Mutual Information Gap (MIG) Chen et al. (2019). The metrics were implemented as proposed in Locatello et al. (2019) with the same hyperparameters. We refer the interested readers to Zaidi et al. (2021) for an intuitive understanding of the metrics. We highlight here that our model achieves a higher DCI metric and a higher corresponding *completeness* and *informativeness* metric, which reflects the mode covering capabilities of the learned representation.

From table 1 we see that on the DSprites dataset, our method outperforms the state of the art models in a majority of the metrics. Similar performance of our model on the 3DShapes dataset as seen in table. Also, the variance in the metrics for the different runs is significantly lower than of the previous methods, thus ensuring a more robust way to disentangle representations. For the real disentanglement dataset of MPI3D Gondal et al. (2019) consisting of a camera taking photos of an object attached to a jointed arm, we see that our model consistently either matches or outperforms the state of the art. From the baselines, especially important is the FactorVAE model, which is the base model upon which we add our contributions for the ProtoVAE model and hence use it as a comparison to demonstrate the effectiveness of our contributions.

On the CelebA dataset, we find that across multiple runs, our model is able to find the same "natural" decompositions that correspond to human-interpretable factors of variation consistently (Fig. 4). We notice that the model is not constrained to completely encode one factor per latent dimension and the model might encode different ranges of a factor in different latents; we see this occur for example when it encodes half of the azimuth in one latent, and half in another. However, as we can see, for the most part, each latent dimension contains information only about one factor of variation and even in the unsupervised regime our model still encodes natural decompositions.

We visualize the embedding space of a trained prototypical network using our method in Fig. 3. We see that when input to the prototypical network is pairs of images from the dataset, with one ground truth factor differing in value between the pair, the prototypical network effectively clusters the pairs based on the differing factor. This clustering aligns with the labels based on the intervened dimensions during training and thus points to the effectiveness of the prototypical network for encouraging disentanglement.

We also performed quantitative and qualitative ablation studies on the 3DShapes dataset by changing the values of $\alpha$, $\lambda$ and $\kappa$ to understand the effectiveness of each of the components and losses we introduce. The results of these ablations can be found in the supplementary material (Appendix C). Furthermore, we also perform ablation studies on the effect of dimension $m$ of the metric space of the prototypical network on the metric scores. We also show in the Appendix (Section C) some limitation cases where the representations of the model did not axis align with a few factors but was rotated with respect to those factors. We see that smaller values of the prototypical network metric space $m$ performs better by encoding data in tighter clusters which in turn it imposes stronger constraints on the VAE. The discriminator and the corresponding $\mathcal{L}_E$ helps in confining the encoding of the factors into a single dimension whereas $\mathcal{L}_P$ alone fails to do this effectively as seen in 5.

## 4 RELATED WORKS

Many state-of-the-art unsupervised disentanglement methods extend the VAE objective function to impose additional constraints on the structure of the latent space to match the assumed independent factor distribution. $\beta$-VAE (Higgins et al., 2017) and AnnealedVAE (Burgess et al., 2018) heavily penalize the KL divergence term thus forcing the learned posterior distribution $q_\phi(z|x)$ to be independent like the prior. Factor-VAE (Kim & Mnih, 2019) and $\beta$-TCVAE (Chen et al., 2019) penalize the total correlation of the aggregated posterior $q_\phi(z)$. $TC = KL(q(z)||\prod_{i=1}^{K} q(z_i))$ where the aggregated posterior is calculated as $q_\phi(z) = \mathbb{E}_{p(x)}[q(z|x_i)] = \frac{1}{N}\sum_{i=1}^{N} q_\phi(z|x_i)$ using adversarial and statistical techniques respectively. DIP-VAE (Kumar et al., 2018) forces the covariance matrix of the aggregated posterior $q(z)$ to be close to the identity matrix by method of moment matching. The changes that we described in the latent space are defined as intervention by Suter et al. (2019) to study the robustness of the learned representations under the Independent Mechanisms (IM) (Schoelkopf et al., 2012) assumption. Most closely related to our work are VAE models that learn to disentangle by altering the latent code. In Jha et al. (2018), the authors use a VAE or AE with a split double latent code, with a cycle consistent loss, but required that attribute labels be known *a priori*, which was also a requirement in Feng et al. (2018) which learned by swapping out chunks of the latent code, and Szabó et al. (2017), which used the labels as a constraint to find unique disentanglement. The authors in Chen et al. (2020) also used cycle-consistent loss, but again required labeling. In Hu et al. (2018) the authors attempted unsupervised disentanglement with regular (non-variational) Autoencoder network models, by stacking one after another, our model instead uses a prototypical neural network. In Lezama (2018) the authors derive a novel Jacobian loss combined with a student-teacher iterative training algorithm with an Autoencoder network model. In Park et al. (2020) the authors develop a latent-manipulating model aimed at human-interactive image manipulation tasks.

Yang et al. (2021) use the group based definition by Higgins et al. (2018) and a cycle consistency loss to define the elements of a group. Our work differs significantly as we do not re-encode the data or the intervention-observations. Zhu et al. (2021) encode the the latent space of a VAE using the commutative Lie group and enforce constraints on the latent space. A recent work Ren et al. (2021) propose to learn disentangled representations from pre-trained models using contrastive methods.

## 5 CONCLUSION AND FUTURE WORK

In this work, we proposed a novel generative model consisting of a VAE and a Prototypical Network for learning disentangled representations in a completely unsupervised way, inspired by recent discovery of sufficient inductive biases. We impose constraints on the structure of the representations learned by training the model in an self-supervised manner to encode information about the different factors in separate dimensions of the representation. Our proposed method is able to outperform other state of the art networks on a number of metrics on three prominent disentanglement datasets. For future work, our method can be easily adapted to be trained in a weakly supervised regime with pairs of data differing in known number of factors being the prototypes for the prototypical network. The results can be possibly improved by intervening on multiple dimensions of the representations simultaneously. The importance of methods that can disentangle data *without labels* is critical as data is plentiful and the resulting representations give interpretable insights into the variations in the

data distribution, and can be used for downstream tasks. Our hope is this work adds evidence that self-supervised generative methods are important in this endeavor.

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

## A  APPENDIX

### A.1  VARIATIONAL AUTOENCODER NETWORKS

Variational Autoencoder networks (VAEs) Kingma & Welling (2014); Rezende et al. (2014) are generative models, which consist of an Inference network and Generator network, instantiated as deep neural networks, which aim to maximize the data log-likelihood calculated according to the generative model $p_\theta(x|z)$ with $z \sim p(z)$:

$$\log p(x) = \log\left( \int_z p_\theta(x|z)\, p(z) dz \right) \tag{12}$$

where $z$ is a lower dimensional latent variable than $x$ and is sampled from a known prior. However, calculating this exact log-likelihood is intractable due to higher dimensional latent space, hence, VAEs instead maximize a lower-bound by approximating the true posterior distribution $p_\theta(z|x)$ with tractable distribution $q_\phi(z|x)$ learned by the Inference network:

$$\max_{\theta,\phi} \mathcal{L}_1(\theta,\phi) = \mathbb{E}_{q(x)}[\mathbb{E}_{q_\phi(z|x)}[\log p_\theta(x|z)] - \beta D_{KL}(q_\phi(z|x)||p(z))] \tag{13}$$

Where $q(x)$ is the empirical distribution of the dataset given by $q(x) = \frac{1}{N}\sum_{i=1}^{N}\delta(x^{(i)})$ and $D_{KL}(q_\phi(z|x)||p(z)) = \mathbb{E}_{q_\phi(z|x)}\log\frac{q_\phi(z|x)}{p(z)}$.

$\theta$ and $\phi$ parameterize the distributions learned by the Generator network and Inference network of a VAE respectively. The top-down generator $p_\theta(x|z)$ and a bottom up Inference network network $q_\phi(z|x)$ and trained jointly to maximize lower bound of the empirical distribution of the training data, when passed through the information bottleneck $z$ Burgess et al. (2018).

Given a training example $x^{(i)} : i \in \{1, \cdots, N\}$, where N is the total number of training points in the dataset, the Inference network network maps $x^{(i)}$ to a lower dimensional representation $z^{(i)}$, which is sampled from the Inference network distribution $q_\phi(z|x)$ ,and the Generator network network maps the representation back to the higher dimensional data space $\hat{x}^{(i)}$. In the objective function in 1, first term measures the difference between the example $x^{(i)}$ and the reconstructed example $\hat{x}^{(i)}$ and the second term measures the distance between the approximate posterior distribution $q_\phi(z|x)$ of the learned representation and the assumed prior distribution $p(z) \sim \mathcal{N}(0, I)$. VAEs, and variants thereof, have shown to disentangle data by imposing structural constraints in the latent space and owing to the information bottleneck principle Burgess et al. (2018). Our network builds on this base capability of the VAE.

## A.2 PROTOTYPICAL NETWORKS

Prototypical Networks Snell et al. (2017), originally used for the few-shot supervised classification regime, are used to learn a metric embedding space where examples belonging to the same class cluster around a prototype representation of the class. The base prototypical network is based around a supervised loss, where the labels are known *a priori* and the network is used to project the data into a new clustered metric space. The network is trained by training episodes consisting of labeled support and query sets, where the support sets are used to find the class prototypes and the query sets are used to evaluate the loss. Mathematically, they are given a labeled support set $S = \{S_1, \cdots S_K\}$, where $K$ is the number of classes, and where each set $S_k = \{(x_k^{(1)}, t_k^{(1)}), \cdots, (x_k^{(N)}, t_k^{(N)})\}$ contains $x_k^{(i)} \in \mathbb{R}^D$ such that $t_k^{(i)} = k$. Prototypical networks compute an $M$-dimensional representation $c_k \in \mathbb{R}^m$ , or *prototype*, of each class, $k \in \{1, \cdots, K\}$, through an embedding function $f_\gamma : \mathbb{R}^D \to \mathbb{R}^m$, usually a deep neural network with parameters $\gamma$.

Each prototype $c_k$ is the mean vector of the embedded support points belonging to its class $k$:

$$c_k = \frac{1}{|S_k|}\sum_{(x_k^{(i)}, t_k^{(i)})\in S_k} f_\gamma(x_k^{(i)}) \tag{14}$$

For a given query point $x_q$, classification is performed by calculating the distance of the query point's embedding $f_\gamma(x_q)$ with the prototypes of the different classes and assigning the label of the closest prototype. While any distance metric $d : \mathbb{R}^M \times \mathbb{R}^M \to [0, +\infty)$ can be used, the Euclidean distance metrics is used in practice which corresponds to the spherical Gaussian density. A distribution over the classes is calculated by applying a softmax function on these distances:

$$p_\gamma(t = k|x_q) = \frac{\exp\left(-d(f_\gamma(x_q), c_k)\right)}{\sum_{k'}\exp\left(-d(f_\gamma(x_q), c_{k'})\right)} \tag{15}$$

The weights of the embedding function are updated by minimizing the negative log-probability of the true class:

$$\min_\gamma \mathcal{L}_2(\gamma) = -\log p_\gamma(t = k|x_q) \tag{16}$$

The above loss is minimized when examples belonging to the same class cluster around the class prototype sufficiently far from the prototypes of the other clusters. For our work, we convert this into an unsupervised training regime by creating labels on the fly in a self-supervised manner, as described in the text.

### A.3 DISENTANGLED REPRESENTATIONS VIA GROUP THEORY

We base our work on the proposed formalized definition proposed by Higgins et al. (2018), which connect the disentangled representations to symmetry transformations using group theory. They define the notion of symmetry transformations as transformations which affect only certain aspects or properties of the data while leaving the rest unchanged. The set of these symmetry transformations constitute a group $G$ and their corresponding effect on some abstract space are defined as the *actions* of the symmetry group $\cdot : G \times X \to X$. Furthermore, they define actions as disentangled with respect to some group decomposition $G = G_1 \times \cdots \times G_n$ if there exist a decomposition of the abstract space $X = X_1 \times \cdots X_n$ and actions $\cdot_i : G_i \times X_i \to X_i$ such that

$$(g_1, \cdots g_n) \cdot (v_1, \cdots v_n) = (g_1 \cdot v_1, \cdots g_n \cdot v_n) \tag{17}$$

Using these definitions Higgins et al. (2018) characterize a representations as disentangled with respect to a particular decomposition of a symmetry group into subgroups if it decomposes into independent subspaces, where each subspace is affected by only the action of a single subgroup.

Importantly, this definition also brings in an *agent* as a critical aspect of the definition. Disentanglement is no longer disembodied here: an agent takes a specific action (a symmetry transformation), and observes the result of the transformation. Formally, we can say that agent's representation of the data is disentangled with respect to a particular group decomposition if: (1) An action $G \times Z \to Z$ exists, (2) there exists a map $f : W \to Z$ which is equivariant between actions on W and Z, and (3) each element of the decomposition of the representation $Z = Z_1 \times ... \times Z_n$ is affected by only one transformation $G_i$.

## B IMPLEMENTATION DETAILS

In this section we give more details and descriptions of our proposed ProtoVAE architecture which consists of. The visual depiction of the model is shown in Fig. 1. The layers and layer sizes are given in Table 2. The hyperparameters used are given in Table 4. The architecture of the latent discriminator a Multi-Layer Perceptron (MLP) with 6 layers with 1000 hidden dimension per layer and 2 output logits as in Kim & Mnih (2019). We normalize the data between [0,1] as input to the Inference network and the output of the Generator network is the mean of a Bernoulli distribution $p_\theta(x|z)$. The reconstruction cost in the first term of 1 is calculated as the negative cross-entropy between the output of the Generator network and the input to the Inference network. Code available at: https://github.com/protovae/protovae.

## B.1 SELF-SUPERVISED DATA GENERATION

---

**Algorithm 1:** Self-Supervised Dataset Creation for Prototypical Network Learning

---

**input** : Data examples $\{x^{(i)}\}_{i=1}^B$, Batch size $B$, latent dimension $d$
**given** : *Inference network: $q_\theta(z|x)$, Generator network: $p_\theta(x|z)$* ;
**output:** Support Set: $S = \{S_1, S_2, ..., S_d\}$ s.t. $S_i = \{s_i^1, \cdots, s_i^B\}$, Query Set:
$\quad\quad Q = \{q_l^1, \cdots, q_l^B\}$, Query set labels $L$, Isometry Loss Labels: $I = \{I_j; \forall j \in [d]\}$

Encode data examples into representation space $\{z^{(i)} \sim q_\theta(z|x^{(i)})\}_{i=1}^B$ with $z \in \mathbb{R}^d$
Generate each Support Set $S_j \ \ \forall j \in d$ :
**for** $j = 1$ **to** $d$ **do**
$\quad\quad \zeta \leftarrow$ random permutation of $[B]$
$\quad\quad z_j^{(i)} \leftarrow z_j^{(\zeta(i))} \quad\quad \forall i \in [B]$: swap the $j^{\text{th}}$ dimension of example $i$ with that of $k_{(k \neq i)}$
$\quad\quad \hat{z}_j^{(i)} \leftarrow (z_1^{(i)}, \cdots z_j^{(i)}, \cdots z_d^{(i)}) \quad \forall i \in [B]$
$\quad\quad \hat{x}_j^{(i)} \sim p_\theta(x|\hat{z}_j^{(i)}) \quad \forall i \in [B]$ generate data from intervened latent
$\quad\quad s_j^{(i)} \leftarrow (\hat{x}^{(i)}, \hat{x}_j^{(i)}) \quad \forall i \in [B]$ assign examples to an element of a support set with label $j$
$\quad\quad S_j \leftarrow \{s_j^1, \cdots, s_j^B\}$
$\quad\quad I_j \leftarrow \{|z^{(i)} - \hat{z}_j^{(i)}| : \quad \forall i \in [B]\}$
**end**
$S \leftarrow \{S_1, \cdots, S_d\}, I \leftarrow \{I_1, \cdots, I_d\}$
Generate Query Set $Q$:
**for** $i = 1$ **to** $B$ **do**
$\quad\quad \chi \leftarrow$ random permutation of $[B]$
$\quad\quad k \in_R [d]$
$\quad\quad z_k^{(i)} \leftarrow z_k^{(\chi(i))}$
$\quad\quad \hat{\mathbf{z}}_k^{(i)} \leftarrow (z_1^{(i)}, \cdots z_k^{(i)}, \cdots z_d^{(i)})$
$\quad\quad \hat{x}_k^{(i)} \sim p_\theta(x|\hat{z}_k^{(i)})$
$\quad\quad q^{(i)} \leftarrow (\hat{x}^{(i)}, \hat{x}_k^{(i)})$
$\quad\quad l^{(i)} \leftarrow k$
**end**
$Q \leftarrow \{q^{(1)}, \cdots, q^{(B)}\}$
$L \leftarrow \{l^{(1)}, \cdots, l^{(B)}\}$

---

## B.2 ARCHITECTURE DETAILS

| INFERENCE NETWORK | GENERATOR NETWORK | PROTONET |
|---|---|---|
| INPUT: CHANNELS X $64 \times 64$ | INPUT: $\in \mathbb{R}^{10}$ | INPUT: 2 X CHANNELS X $64 \times 64$ |
| $4 \times 4$ CONV. 32 LRELU. STRIDE 2 | FC. 128 LRELU. | $4 \times 4$ CONV. 32 LRELU. STRIDE 2 |
| $4 \times 4$ CONV. 32 LRELU. STRIDE 2 | FC. $4 \times 4 \times 64$ LRELU. | $4 \times 4$ CONV. 32 LRELU. STRIDE 2 |
| $4 \times 4$ CONV. 64 LRELU. STRIDE 2 | $4 \times 4$ UPCONV. 64 LRELU. STRIDE 2 | $4 \times 4$ CONV. 64 LRELU. STRIDE 2 |
| $4 \times 4$ CONV. 64 LRELU. STRIDE 2 | $4 \times 4$ UPCONV. 64 LRELU. STRIDE 2 | $4 \times 4$ CONV. 64 LRELU. STRIDE 2 |
| FC. 128 LRELU. | $4 \times 4$ UPCONV. 64 LRELU. STRIDE 2 | FC. 128 LRELU. |
| FC. $2 \times 10$. | $4 \times 4$ UPCONV. 1 SIGMOID. STRIDE 2 | FC. 2 , FC. 10 |

Table 2: Network Architecture of the VAE Inference network, VAE Generator network, and the Prototypical Network.

## B.3 NETWORK HYPERPARAMETERS

We detail the hyperparameters used for each dataset. Table 4 shows the full list of values for each component.

| PARAMETER (DSPRITES) | VALUE | PARAMETER (3DSHAPES, CELEBA) | VALUE |
| --- | --- | --- | --- |
| LATENT SPACE DIMENSION | 10 | LATENT SPACE DIMENSION | 10 |
| METRIC SPACE DIMENSION | 3 | METRIC SPACE DIMENSION | 2 |
| OPTIMIZER | ADAM | OPTIMIZER | ADAM |
| OPTIMIZER (BETA1) | 0.9 | OPTIMIZER (BETA1) | 0.9 |
| OPTIMIZER (BETA2) | 0.999 | OPTIMIZER (BETA2) | 0.999 |
| LEARNING RATE (VAE) | 1e-4 | LEARNING RATE (VAE) | 1e-04 |
| LEARNING RATE (GAN) | 1e-4 | LEARNING RATE (GAN) | 5e-05 |
| LEARNING RATE (PROTO) | 1e-4 | LEARNING RATE (PROTO) | 1e-04 |
| BATCH SIZE | 128 | BATCH SIZE | 128 |
| EPOCHS (TRAIN) | 50 | EPOCHS (TRAIN) | 30 |
| $\alpha$ (TOTAL CORRELATION (GAN) | 10 | $\alpha$ (TOTAL CORRELATION (GAN) | 20 |
| $\lambda$ (PROTOTYPICAL LOSS) | 10 | $\lambda$ (PROTOTYPICAL LOSS) | 20 |
| $\kappa$ (ISOMETRY LOSS) | 10 | $\kappa$ (ISOMETRY LOSS) | 20 |

Table 3: Hyperparameters values for the different components of the network for the different datasets.

| PARAMETER (MPI3D) | VALUE |
| --- | --- |
| LATENT SPACE DIMENSION | 10 |
| METRIC SPACE DIMENSION | 3 |
| OPTIMIZER | ADAM |
| OPTIMIZER (BETA1) | 0.9 |
| OPTIMIZER (BETA2) | 0.999 |
| LEARNING RATE (VAE) | 1e-4 |
| LEARNING RATE (GAN) | 5e-5 |
| LEARNING RATE (PROTO) | 1e-4 |
| BATCH SIZE | 128 |
| EPOCHS (TRAIN) | 30 |
| $\alpha$ (TOTAL CORRELATION (GAN) | 10 |
| $\lambda$ (PROTOTYPICAL LOSS) | 2 |
| $\kappa$ (ISOMETRY LOSS) | 2 |

Table 4: Hyperparameters values for the different components of the network for the MPI3D dataset

## C   ABLATION STUDIES (METRICS AND IMAGES)

We demonstrate qualitative and quantitative results showing the effectiveness of the different hyperparameters for disentanglement by varying them across a range while keeping the others the same. We perform these ablations studies on the 3D shapes dataset with hyperparameters given in Table 4 except the one hyperparameter which is varied. in Table 5 we vary the dimension of the metric space of the prototypical network and see it's effect on the various metrics. In Table 5 we vary the strength of the total correlation loss as measured by the latent space GAN to train the VAE by varying the hyperparameter $\alpha$. In Table 6 we perform similar experiments by varying the hyperparameter $\lambda$ which scales the combined loss of the consistency and uniqueness loss and the regularization strength of the Isometry loss which is governed by the hyperparameter $\kappa$.

Qualitatively we show the latent traversals of results where learned latent representations are not axis aligned with the ground-truth factors of variations. In Figure 7 we show the effects of having a lower value of $\lambda$ and see that even though each dimension encodes information related to one factor, the complete range of the factor is encoded by multiple dimensions. In Figure 8 we show the effect of having a lower regularization for the Isometry loss ($\kappa$) and see that we learn a subset of factors which are rotated in space and are not axis aligned with the ground truth factors.

In addition, we perform an ablation study by removing the GAN discriminator and seeing the result of just the VAE and prototypical network in Fig. 9. We found that without the GAN, a lower dimensional prototype metric space is needed (we use dimension $d = 1$), but that the network is

| METRIC | $m = 2$ | $m = 5$ | $m = 8$ | METRIC | $\alpha = 0$ | $\alpha = 10$ | $\alpha = 20$ |
|--------|---------|---------|---------|--------|--------------|---------------|---------------|
| FVAE | $.90 \pm .06$ | $.88 \pm .04$ | $.87 \pm .04$ | FVAE | $.93 \pm .04$ | $.85 \pm .03$ | $.88 \pm .05$ |
| DCI | $.84 \pm .07$ | $.77 \pm .05$ | $.74 \pm .05$ | DCI | $.78 \pm .05$ | $.81 \pm .06$ | $.81 \pm .07$ |
| MIG | $.71 \pm .11$ | $.62 \pm .08$ | $.57 \pm .06$ | MIG | $.59 \pm .07$ | $.63 \pm .04$ | $.65 \pm .08$ |
| $\beta$-VAE | $.92 \pm .05$ | $.91 \pm .05$ | $.91 \pm .04$ | $\beta$-VAE | $.92 \pm .06$ | $.88 \pm .04$ | $.90 \pm .05$ |

Table 5: Ablation study showing the performance on different metrics for our proposed ProtoVAE, with different values of the prototype dimension space $m$ (*left*) and $\alpha$ (*right*) on the 3D shapes dataset. In some instances, these versions outperform the results posted in the main paper. We see that smaller values of the prototypical network dimension $d$ actually perform better as it imposes stronger constraints on the VAE. We also see that $\alpha = 0$ performs well for the $\beta$ VAE and the FactorVAE metrics but does not perform as well on the MIG and the DCI scores as the Latent Space GAN helps in confining the encoding of the factors into a single dimension whereas our losses do not do this as effectively.

| METRIC | $\kappa = 0$ | $\kappa = 10$ | $\kappa = 40$ | METRIC | $\lambda = 0$ | $\lambda = 10$ | $\lambda = 20$ |
|--------|--------------|---------------|---------------|--------|---------------|----------------|----------------|
| FVAE | $.68 \pm .04$ | $.84 \pm .03$ | $.72 \pm .04$ | FVAE | $.84 \pm .06$ | $.82 \pm .04$ | $.88 \pm .05$ |
| DCI | $.57 \pm .06$ | $.72 \pm .04$ | $.69 \pm .05$ | DCI | $.64 \pm .05$ | $.71 \pm .02$ | $.81 \pm .07$ |
| MIG | $.43 \pm .06$ | $.56 \pm .08$ | $.58 \pm .08$ | MIG | $.58 \pm .06$ | $.57 \pm .06$ | $.65 \pm .08$ |
| $\beta$-VAE | $.80 \pm .05$ | $.84 \pm .06$ | $.88 \pm .06$ | $\beta$-VAE | $.88 \pm .04$ | $.88 \pm .05$ | $.90 \pm .05$ |

Table 6: Ablation study showing the performance on different metrics for our proposed ProtoVAE, with different values of the hyperparameter $\kappa$ (*left*) and $\lambda$ (*right*) on the 3D shapes dataset.

still able to disentangle well. We leave for future work more investigations into removing the GAN component entirely.

Another promising future work would be to impose constraints on the structure of the metric space to obtain more separable clusters. Moreover, curriculum learning could be used while performing interventions to ensure that the intervened dimension is changed by a significant value and then to slowly decrease this value as training proceeds.

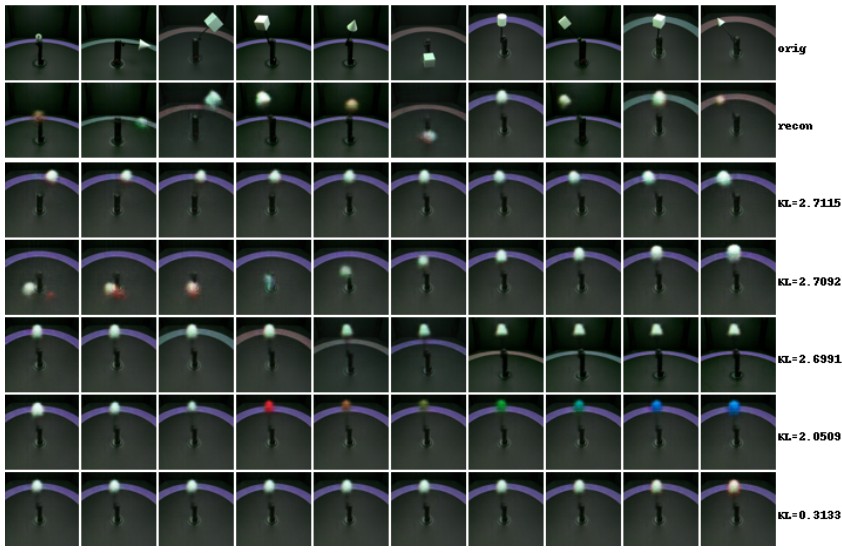

Figure 5: Latent traversals on the MPI3D real world disentanglement dataset. The data is collected via a camera that observes a jointed arm with known changed ground truth factors of variation. From top to bottom: original data, reconstruction, arm angle left/right, arm angle top/bottom, background height, arm end color, size.

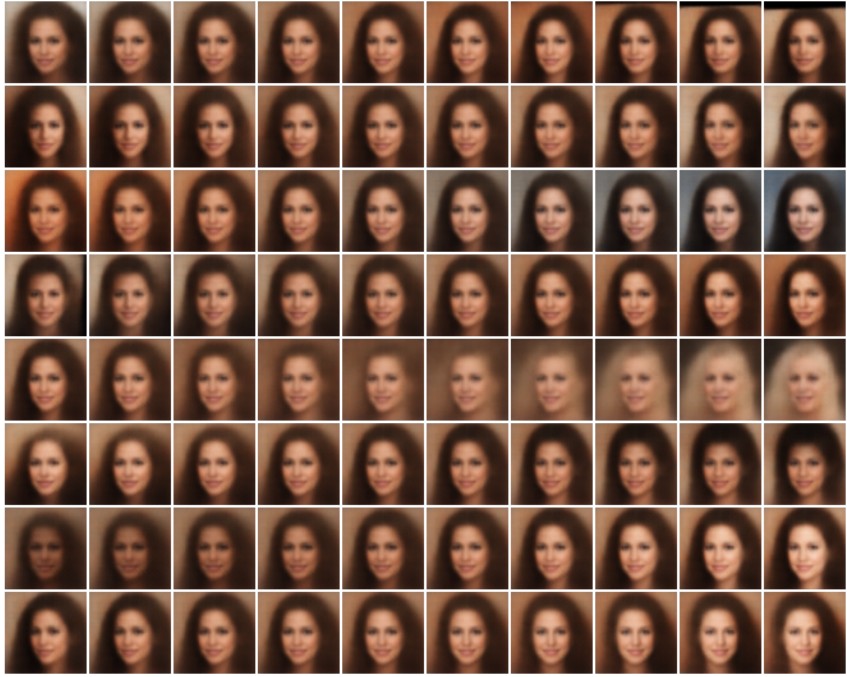

Figure 6: More latent traversals on the CelebA dataset. We see that the traversals are similar to the ones discovered in Fig. 4 of the main paper.

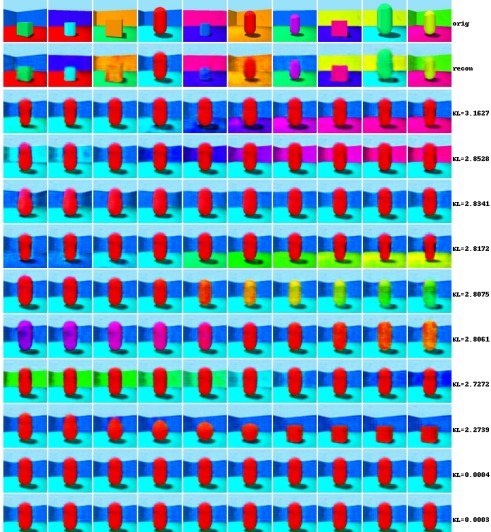

Figure 7: Results on *3dShapes* with a lower value of $\lambda$, showing theoretically correct disentanglement that disagrees with a "natural" decomposition - namely, that the factors need not be completely capture by one latent, and may be split into several; for example, here, color change need not cover the entire spectrum in one traversal. In row 7, 8, *left* and row 6,8 *right*, we can see that color traversal was split into two latents. This still follows the group theoretical definition of disentangled representations. And given a high value of $\kappa$, the mapping between the dimensions and the corresponding traversals is isometric.

## C.1    PROTOTYPICAL NETWORK EMBEDDINGS

We take a closer look at the workings of the prototypical network component, and analyze its output. To do so, we trained a network using prototypical metric dimension $d = 2$, so that we can plot the

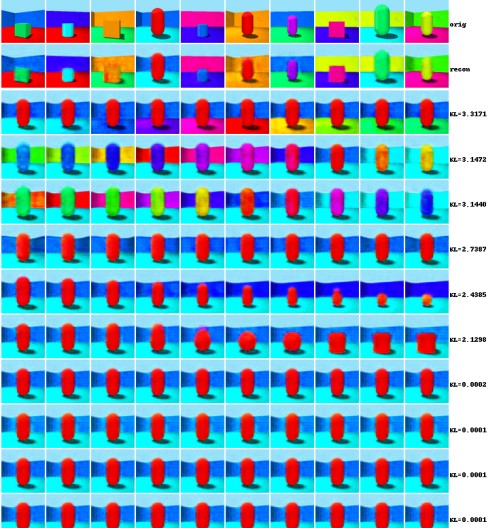

Figure 8: Similar to Fig. 7, a non-unique decomposition can be found by a 'rotation' of that factors such that facets of each factor are mixed together. We can see this here with the wall and shape color being intermingled. For this run, had a lower value of the Isometry loss scaling ($\kappa$). This shows that this loss was important in ensuring a more natural decomposition.

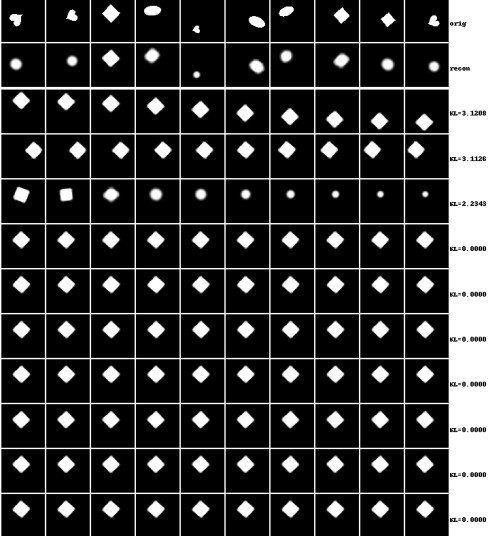

Figure 9: Results of the ablation study removing the GAN discriminator and just training a VAE with the prototypical network, on a low dimensional prototypical network metric space $d = 1$; we find that removing the GAN requires a lower dimensional metric space. However, even without the GAN discriminator we find that the network performs remarkably well at disentangling the factors of variation into natural decompositions. We also found that we needed higher values of $\lambda$ and $\kappa$ for better disentanglement. Due to this however, with the GAN removed not all factors are captured and the reconstructions are slightly blurrier. We leave to future work further development of this variant without the GAN discriminator.

resulting points in 2D space. We pass into the prototypical network network two data samples from the *3DShapes* dataset that differ in one ground-truth factor of variation (mimicking the intervention process, but with ground truth examples), and plotted the corresponding projection into the $\mathbb{R}^2$ space. This tests to see if the trained prototypical network will generalize the clustering to ground truth pairs of data which differ in only one factor as shown in Fig. 12. If the prototypical network

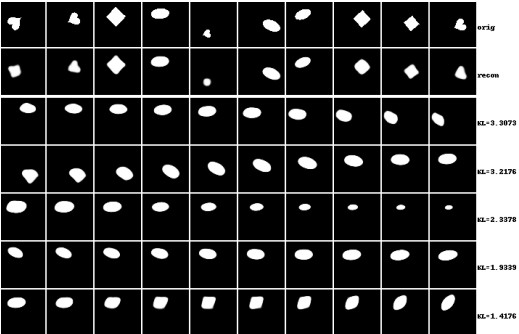

Figure 10: Results of latent traversals for the DSprites dataset for two baseline methods of $\beta$-VAE and FactorVAE. *left* The latent traversals of representations learned by a $\beta$-VAE with the same hyperparameters as listed in 4 and $\beta = 4$. *right* The latent traversals for the FactorVAE model with $\gamma = 40$. We see that our method qualitatively outperforms both these methods on the DSprites dataset with better reconstruction quality.

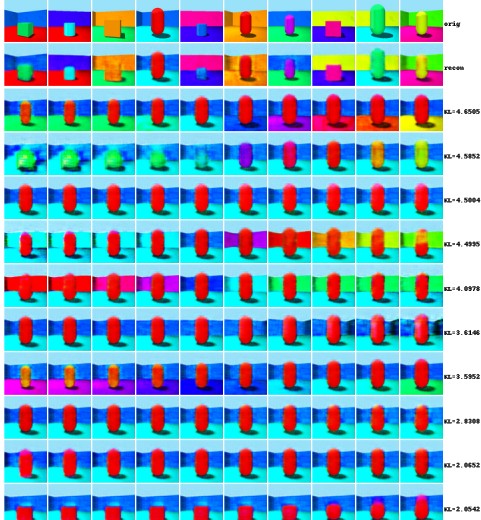

Figure 11: Results of latent traversals for the 3Dshapes dataset for the baseline method of Factor-VAE. *left* The latent traversals of representations learned by the FactorVAE for $\gamma = 20$ with the same hyperparameters as listed in 4. *right* The latent traversals for the FactorVAE model with $\gamma = 40$. We see that our method qualitatively outperforms both these methods on the 3Dshapes dataset with better reconstruction quality and more disentangled factors encoded in the different latents.

has been trained correctly and aids in disentanglement of the factors, it would cluster changes to the same latent in similar clusters in $\mathbb{R}^2$. Notably, here we supply the prototypical network directly with *two real dataset images*, which is not the same distribution that the prototypical network is trained on. During training the prototypical network is trained on a pair of one reconstruction of a real sample and one intervention-observation. Furthermore, we also see the effect of the regularization effect of the $\lambda$ parameter on the clustering for a model with the other parameters the same. In Fig. 3, we see the clusters formed by a trained model on the 3D shapes dataset.

This suggests that during training the prototypical network clusters the pair of reconstructed and intervention-observation according to the index of the dimension of the representation that was intervened on to create that pair. This in turn forces the Generator network of the VAE to make consistent and unique changes to the data based on the intervened dimension. Furthermore, it is apparent that clusters of some factors are well separated whereas clusters are closer in the metric space based on the entity that they affect.

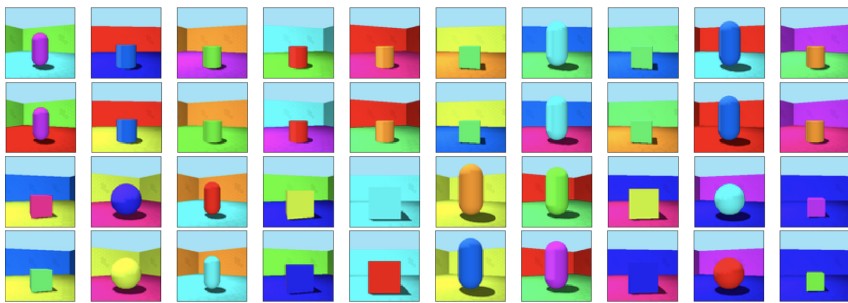

Figure 12: Inputs to the prototypical network. The first two rows represent pairs (each pair is a column) where the floor hue is the only factor that is differing amongst the pair of images. Similarly the bottom two rows represent pairs where the object hue is the differing factor.

## D    COMPARISON OF MODELS WITH FEW KNOWN FACTORS

To show the effectiveness of the Prototypical network in disentangling the factors of variation we perform ablation studies with smaller number of factors of variation, with the latent space being equal to the number of factors. We show the comparison of our method with the $\beta$-VAE and the Factor-VAE that we trained ourselves for fairness of comparison. As an ablation, we created subset of synthetic datasets from DSprites containing only two factors of variation: one with only position x and position y, and one with the orientation (o) and position y factors of variations. We show the reconstructions and the latent traversals of our method in the in Figure **??** and Figure **??** for the position x and position y factors and the orientation and position y factors respectively. From the figure we can see that the ProtoVAE effectively disentangles the different factors into separate dimensions of the latent representation. We can see that for the subdataset consisting of x and y, the FactorVAE and $\beta$-VAE correctly disentangle the two factors (Fig. **??** and Fig. **??**), as does our proposed ProtoVAE. However, with y and orientation, we see that the FactorVAE (Fig. **??**) and the $\beta$-VAE (Fig. **??** do not fully "naturally" disentangle the factors, leading to mixed representations in the latents.

For this, we used a prototypical metric dimension of $d = 1$. We found that the GAN is only necessary for larger metric dimension space, such as the ones we use for the main paper results. In Figure **??** and Figure **??** we show the Factor-VAEs performance and in Figure **??** and Figure **??** we show the $\beta$-VAEs performance on the same datasets. We see that the ProtoVAE outperforms these methods.

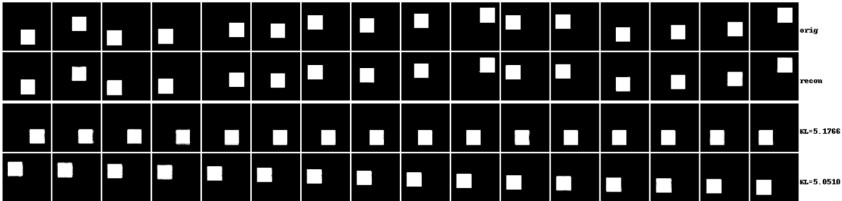

Figure 13: ProtoVAE (x & y):Traversal results for $d = 1$ ($\alpha = 0, \lambda = 10, \kappa = 10$)without the latent discriminator, on the sub-dataset from DSprites created by only varying x and y (same shape, orientation, and size.) The first row consists of examples from the dataset, the second row consists of the corresponding reconstructions. The third row shows the latent traversals for the $1^{\text{st}}$ dimensions which encodes the position x factor and the $2^{\text{nd}}$ row which encodes the position y factor.

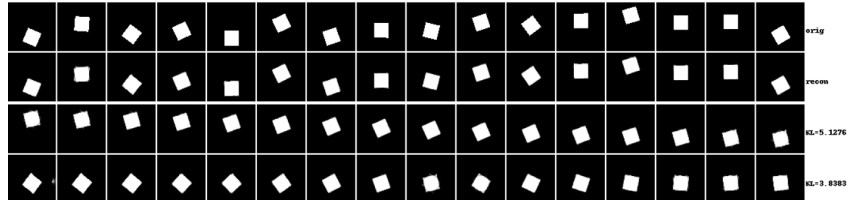

Figure 14: ProtoVAE (orientation & y-position)Traversal results for $d = 1$ without the latent discriminator, on the sub-dataset from DSprites created by only varying y and orientation (same shape, position x, and size.) The third row shows that the dimension 1 of the latent representation encodes the position y factor and dimension 2 encodes the orientation factor. Unlike FactorVAE and $\beta$-VAE (Fig. **??**, Fig. **??**), our proposed model fully disentangles the factors into separate latents without rotation.

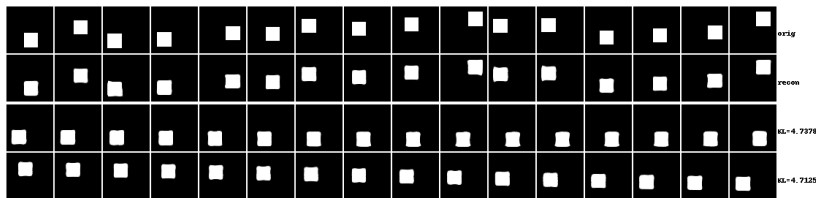

Figure 15: FactorVAE (x position & y position): Latent traversal results for FactorVAE with hyperparameter value 10. The FactorVAE correctly disentangles these two factors without any mixing.

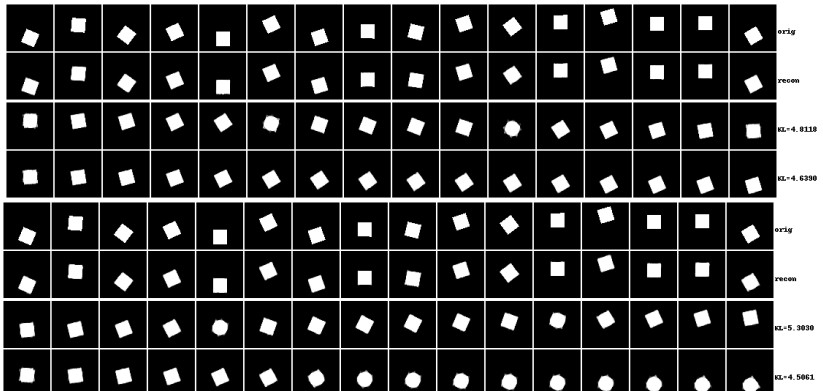

Figure 16: FactorVAE (orietnation & y-position): Traversal results for FactorVAE on the orientation and y-position dataset, showing only partial disentanglement, or a rotation of the factors. *Top four rows*: Hyperparameter value for the total correlation loss is 5. *Bottom four rows*: The value of the hyperparameter is 10. We see that a rotational subspace of the factors is learned. In particular, we can see in row 3 *top*, the y position and orientation is mixed.

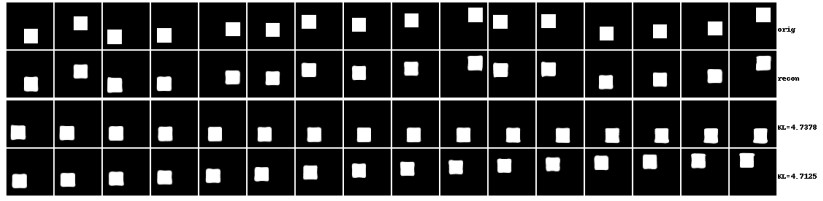

Figure 17: $\beta$-VAE on the subdataset of (x-position & y-position): Traversal results with $\beta = 10$ We see that the $\beta$-VAE did train successfully to disentangle x and y.

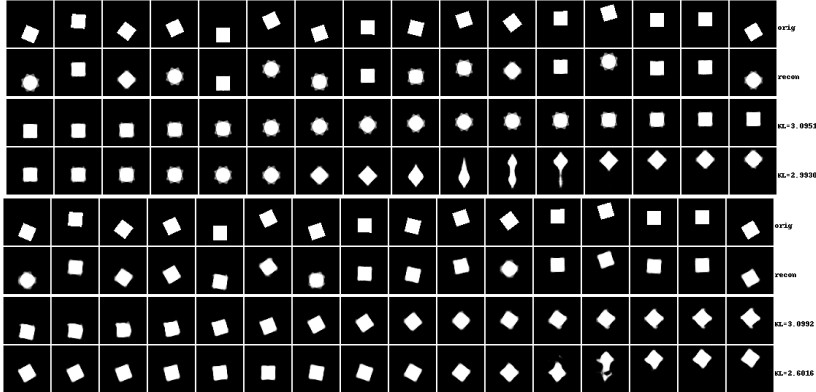

Figure 18: $\beta$-VAE on orientation and y-position: Traversal results for $\beta$-VAE on the orientation and position y dataset. *Top four rows*: $\beta = 5$. *Bottom four rows*: $\beta = 10$. We can see the network performs more poorly on these factors than our proposed model (row 4, 8) in generating smooth traversals.

