# OpenReview forum: "ProtoVAE: Using Prototypical Networks for Unsupervised Disentanglement"
_ICLR.cc/2023/Conference — Submitted to ICLR 2023_

### Official Review · Reviewer_L6Gs · 2022-10-21

**Confidence:** 4
**Correctness:** 1
**Technical Novelty And Significance:** 1
**Empirical Novelty And Significance:** 2
**Recommendation:** 3

**Clarity, Quality, Novelty And Reproducibility:**

- Many parts of this work are based on verbal explanations without mathematical formulation. Some terms are not used in their more common sense in the context of disentanglement, such as intervention (causal inference) and action (group theory).
- Several parts of the proposed method are not technically sound. For example, Eq. (3) uses a product of log-likelihood and KL divergence without any explanation.
- It is hard to understand how to implement each part. The author did not provide the code, either.


**Strength And Weaknesses:**

The author attempted to solve an important and challenging problem -- unsupervised disentanglement -- but seems to have several fundamental misunderstandings of existing theories and tools:

- The author misinterpreted the result of [Locatello et al., 2019] and claimed that unsupervised disentanglement is possible *as long as there is a proper inductive bias*. However, the author did not mathematically prove the proposed method can actually lead to disentanglement.
- It seems the author does not know the difference between VAE and beta-VAE. They cited the original VAE paper (well there is even no citation in Section 2.2) but used beta-VAE instead, showing that they are unaware of the basic derivation of the learning objective (log-likelihood vs. information bottleneck).
- The method is overcomplicated. It is unclear which part is essential and which part is optional. For example, the effect of "latent space GAN" is confusing.

**Summary Of The Paper:**

This paper proposed to combine variational autoencoder, prototypical network, and generative adversarial network in a way to solve unsupervised disentanglement. The proposed method was evaluated only empirically.

**Summary Of The Review:**

This work tried to solve a challenging problem but did not properly use existing methods and theories. The author misinterpreted and miscited some existing works. The mechanism of the proposed method is unclear due to the confusing writing style of this paper. Therefore I'm afraid I  cannot recommend acceptance of this paper to ICLR.

---

> ### Author Response · Authors · 2022-11-17
> **Reply to Reviewer L6Gs**
>
> We thank the reviewer for the feedback and hope to use it to improve the paper.  We have a few replies to some of the comments that the reviewer made:
>
> *The author misinterpreted the result of [Locatello et al., 2019] and claimed that unsupervised disentanglement is possible as long as there is a proper inductive bias. However, the author did not mathematically prove the proposed method can actually lead to disentanglement*: We interpreted the result such that the converse of the claim would also be true and mentioned the converse instead. Local isometry can serve as a good inductive bias was claimed in horan et al. We refer you to that paper for a detailed explanation. Moreover, we do not claim to mathematically prove it but show it empirically.
>
> *It seems the author does not know the difference between VAE and beta-VAE. They cited the original VAE paper (well there is even no citation in Section 2.2) but used beta-VAE instead, showing that they are unaware of the basic derivation of the learning objective (log-likelihood vs. information bottleneck)*: We use a generalized form for both the VAE and beta-VAE, where the beta-VAE is different from the original VAE by a multiplicative factor in the KL-divergence loss.  The original VAE objective is equivalent to the beta-VAE with a beta factor of 1. We have amended the text to make this clearer.
>
> *The method is overcomplicated. It is unclear which part is essential and which part is optional. For example, the effect of "latent space GAN" is confusing*: We used the latent space GAN to refer both to the discriminator and the inference network together. This discriminator is necessary to confine a factor in a single latent as some of the metrics report higher scores if this condition is met. Though it is hugely debated whether this is necessary for practical disentanglement.  We have tried to explain this more thoroughly in the new version.
>
> *Many parts of this work are based on verbal explanations without mathematical formulation. Some terms are not used in their more common sense in the context of disentanglement, such as intervention (causal inference) and action (group theory)*: We have used the word intervention in the same sense as [1]. To address the confusion with the word “action”, we have removed it from the paper.
>
> *Several parts of the proposed method are not technically sound. For example, Eq. (3) uses a product of log-likelihood and KL divergence without any explanation*: We have added an explanation for this product. We multiply by the KL loss so that the dimensions which do not encode any information (higher KL values mean more information is encoded) do not contribute to the losses.
>
>
> We thank the reviewer for the feedback and suggestions and tried to incorporate the suggestions into the new draft.
>
> [1] Suter, R., Miladinovic, D., Schölkopf, B., & Bauer, S. (2019, May). Robustly disentangled causal mechanisms: Validating deep representations for interventional robustness. In International Conference on Machine Learning (pp. 6056-6065). PMLR.
> Chicago

---

### Official Review · Reviewer_69qY · 2022-10-23

**Confidence:** 4
**Correctness:** 3
**Technical Novelty And Significance:** 3
**Empirical Novelty And Significance:** 1
**Recommendation:** 3

**Clarity, Quality, Novelty And Reproducibility:**


Clarity, quality, and reproducibility: as described above, the paper needs improvement in these aspects.

Novelty: As far as I know, the idea of incorporating a prototypical network in training disentangled VAEs has not been explored before and is novel.

**Strength And Weaknesses:**


Strength:

* The idea is technically sound.

Weaknesses:

* The writing needs a lot of improvement. The main text is not self-contained: some important technical details and definitions of notations are deferred to the appendix. In addition, even after reading both the main text and the appendix, some important details are still unclear to me. Also, there are many typos. Detailed questions are below.

    - Page 2: Given that locally isometric is the motivation of one key component of the proposed algorithm, it is important to explain what it means instead of only mentioning the term.
    - Page 2: what does "the interpretability of GANs" mean here?
    - Page 3: equation 13 should be equation 2
    - Page 3: The core algorithm of generating samples for prototypical networks is deferred to the appendix. The important notations that the following paper depends on (e.g., support set, query set) are also in the appendix and are not defined anywhere in the main text.
    - Page 3: "The the" should be "The"
    - The notations are not consistent throughout the paper: z and \hat{z}_k are sometimes in bold and sometimes not. The proposed algorithm is named ProtoVAE or Proto-VAE in different places.
    - Page 4: Again, the important details of the prototypical network (needed for understanding the paper) are deferred to Appendix A.2.
    - Equation 3: I understand the KL coefficient is used for scaling down the loss when the dimension is uninformative. However, why do you choose KL instead of other distance metrics? This is not explained in the paper.
    - Equation 6 does not match what is described in the text "the prototypes are actions of the different dimensions on a particular example and the query example is the action of a randomly chosen dimension on that example". Because of that, I am not able to check the correctness/soundness of this part of the algorithm.
    - Page 6: "To ensure that interventions in these dimensions do not contribute to the losses, we scale equations 5 and 3 with the KL divergence values of the intervened latent dimension." is already explained before. Maybe you can remove it.
    - Appendix B.1: Is "k_{(k\not= i)}" a typo? k is not used in this line.

* The scores of the proposed approach are much worse than state-of-the-art, and the baseline scores reported in the paper are worse than what was reported in prior work. Details are below.

    - Most of the reported scores of baselines are worse than what was reported in prior work (e.g., https://arxiv.org/pdf/1906.06034.pdf). More importantly, the proposed ProtoVAE is built upon FactorVAE, but the ProtoVAE's scores are worse than FactorVAE's scores reported in prior work (e.g., https://arxiv.org/pdf/1802.05983.pdf).
    - I understand that different implementations may lead to different scores. As the paper states, the hyperparameter settings and experimental conditions are taken from Locatello et al. But the reported scores are still not consistent with what was reported in Locatello et al.

    Therefore, I am not convinced by the experimental results and have concerns about how much ProtoVAE improves upon prior work.


**Summary Of The Paper:**

The paper proposes a new approach for training disentangled variational autoencoder (VAE). The approach builds on the top of FactorVAE, and adds a prototypical network that clusters pairs of generated samples that differ in one latent dimension. The encoder and decoder are trained to be easy to be classified by the prototypical network, thus encouraging a disentangled latent space. Experiments are conducted on dSprites, 3DShapes, MPI3D, and CelebA datasets.

**Summary Of The Review:**


Overall, the idea is interesting, but the current writing quality and experimental results are below the threshold. I would recommend the authors polish the writing and add more clarifications on the experimental results.

---

> ### Author Response · Authors · 2022-11-17
> **Reply to Reviewer 69qY**
>
> We thank the reviewer for their thorough feedback and helpful comments, and have revised our submission throughout to address the important comments the reviewer has made.  In general, we have tried to address the confusion about the model components by adding a clearer description of the model overview, with connections to the diagram, which we hope elucidates the model structure.  We also would like to point out a few specific replies to the reviewers’ questions:
>
> *equation 13 should be equation 2?*: Fixed
>
> *The core algorithm of generating samples for prototypical networks is deferred to the appendix. The important notations that the following paper depends on (e.g., support set, query set) are also in the appendix and are not defined anywhere in the main text*: We have made the change in Section 2.
>
> *Page 3: "The the" should be "The"* : Fixed
>
> *The notations are not consistent throughout the paper: z and \hat{z}_k are sometimes in bold and sometimes not. The proposed algorithm is named ProtoVAE or Proto-VAE in different places* : Fixed
>
> *Equation 3: I understand the KL coefficient is used for scaling down the loss when the dimension is uninformative. However, why do you choose KL instead of other distance metrics? This is not explained in the paper*: We multiply by the KL loss so that the dimensions which do not encode any information (higher KL values mean more information is encoded) do not contribute to the losses. The KL divergence signifies the divergence of the posterior distribution from the uninformative prior. Moreover, in the VAE training phase the KL divergence is already calculated.
>
> *Equation 6 does not match what is described in the text "the prototypes are actions of the different dimensions on a particular example and the query example is the action of a randomly chosen dimension on that example". Because of that, I am not able to check the correctness/soundness of this part of the algorithm*: We have added more explanation for the equation. Hopefully the writing now makes this clearer.
>
> *Page 6: "To ensure that interventions in these dimensions do not contribute to the losses, we scale equations 5 and 3 with the KL divergence values of the intervened latent dimension." is already explained before. Maybe you can remove it*: Fixed
>
> *Appendix B.1: Is "k_{(k\not= i)}" a typo? k is not used in this line* : This was used just to explain the mechanism of swapping the values in the dimension across a batch. The comments have been added to improve the readability of the algorithm.
>
> *The scores of the proposed approach are much worse than state-of-the-art, and the baseline scores reported in the paper are worse than what was reported in prior work*: The reasons are below:
> 1. Most of the papers [1,2] only reported their highest runs for specific architectures and after a lot of hyperparameter tuning. We instead report the mean and the standard deviation (std) after averaging over 10 runs with different random seeds ([3])
> 2. Results might not be exactly the same as [3] as they averaged over 50 runs whereas we averaged over 10 runs. However our means are within one std of theirs.
> 3. The recent papers that have attempted to reproduce their results report similar scores as we have [4,5].
>
> We hope the revised draft addressed all the reviewers’ concerns, and questions, and look forward to any more feedback the reviewer may have. We thank you for taking the time to review our paper.
>
> [1] Kim, H., & Mnih, A. (2018, July). Disentangling by factorising. In International Conference on Machine Learning (pp. 2649-2658). PMLR.
> [2] Chen, R. T., Li, X., Grosse, R. B., & Duvenaud, D. K. (2018). Isolating sources of disentanglement in variational autoencoders. Advances in neural information processing systems, 31.
> [3] Locatello, F., Bauer, S., Lucic, M., Raetsch, G., Gelly, S., Schölkopf, B., & Bachem, O. (2019, May). Challenging common assumptions in the unsupervised learning of disentangled representations. In international conference on machine learning (pp. 4114-4124). PMLR.
> [4] Ren, X., Yang, T., Wang, Y., & Zeng, W. (2021, September). Learning disentangled representation by exploiting pretrained generative models: A contrastive learning view. In International Conference on Learning Representations.
> [5] Yang, T., Ren, X., Wang, Y., Zeng, W., Zheng, N., & Ren, P. (2021). GroupifyVAE: From group-based definition to VAE-based unsupervised representation disentanglement. arXiv preprint arXiv:2102.10303.

---

> > ### Comment · Reviewer_69qY · 2022-11-21
> > **Thank you!**
> >
> > Thank the authors for the detailed answer. Because the (requested) changes are significant and my concerns about the significance of the results remain, I will keep the score.

---

### Official Review · Reviewer_Fvdi · 2022-10-24

**Confidence:** 4
**Correctness:** 3
**Technical Novelty And Significance:** 3
**Empirical Novelty And Significance:** 3
**Recommendation:** 3

**Clarity, Quality, Novelty And Reproducibility:**

There are considerable issues with the clarity of this paper, to the extent that actually trying to decipher the model being proposed is extremely hard. Section 2 attempts, but fails, to clearly explain the model and all the design decisions. It is rather difficult to decipher where the actual contributions in the model design come from as they are poorly explained. Too much space is devoted to explaining existing models (VAE or actually Beta-VAE in Section 2.2, prototypical nets in 2.4, etc). One has to refer to the algorithmic description in the appendix to try and understand section 2.3 - a plain English description of how the model works and why is missing.

Further to these points, the technical writing is of quite low quality and littered with errors, such as duplicate equations and wrong cross-referencing. In Algorithm 1 there are considerable issues of clarity and undefined variables (what is $k$ for example - in the second for loop its clear, but it appears in the comments of the first loop).

In terms of reproducibility, it might be possible to reproduce the results with trial and error to try and get all the architectural components to match, but this would likely be a significant undertaking.

**Strength And Weaknesses:**

### Strengths

- Experimental results appear to show good results. An appropriate range of measures is used, and the method is compared against a range of other approaches.
- I _think_ the overall idea is sound, and is novel

### Weaknesses

- Clarity and presentation leaves a lot to be desired (see below)
- Many of the experimental aspects are rather unclear
- It's not clear to me that the "latent space GAN" is actually a GAN; is it not just a discriminator network that's used in the loss?


**Summary Of The Paper:**

This paper proposes an unsupervised approach to learning disentangled latent factors within a VAE-like model. The core idea of the approach appears to be the swapping of a single value within the latent code between batch items during training; the decoded original latent and latent with the swapped value can then be compared allowing the network to isolate specific changes in the data space that the swapping induces. The training proceedusre forces the model to learn to ensure that each single latent dimension only changes one factor in the data space. Experimental results seem to indicate that the proposed approach does a significantly better job at disentanglement than existing approaches when compared using a range of measures.


**Summary Of The Review:**

As I said in the strengths section, I do think that there might be a good, novel, idea buried within this paper. However as currently written that idea is not clear enough to justify acceptance. Hopefully the authors will work on this aspect. In terms of scoring for correctness, novely and significance I've tried to be positive, scoring on what I think the paper is trying to say rather than necessarily what it does say.

---

> ### Author Response · Authors · 2022-11-17
> **Reply to Reviewer Fvdi**
>
> We thank the reviewer for their thoughtful comments and recommendations.  We have thoroughly revised our submission draft to accommodate the suggestions the reviewer gave.  Specifically, we have attempted to dramatically increase the readability and improve the model description section. We also would like to give a few replies to some of the questions raised:
>
> *It's not clear to me that the "latent space GAN" is actually a GAN; is it not just a discriminator network that's used in the loss?​*​: We have renamed this component a discriminator network in the text to make it clear it uses the encoder as the generator component.  We hope the new version is more understandable.
>
> *There are considerable issues with the clarity of this paper...*: Section 2 has been updated to reflect these suggestions and feedback - we refer the reviewer there.
>
> *Further to these points, the technical writing is of quite low quality and littered with errors, such as duplicate equations and wrong cross-referencing. In Algorithm 1 there are considerable issues of clarity and undefined variables (what is for example - in the second for loop its clear, but it appears in the comments of the first loop)*: While they look similar, the subtle differences in the equations are actually very important. We tried to make changes to make this clearer. In the algorithm the comments were added to improve readability (k was specifically added to explain the swapping mechanism)
>
> We hope our new draft has sufficiently addressed the reviewers’ concerns, and look forward to any more feedback.

---

### Official Review · Reviewer_jevi · 2022-10-28

**Confidence:** 3
**Correctness:** 2
**Technical Novelty And Significance:** 2
**Empirical Novelty And Significance:** 2
**Recommendation:** 3

**Clarity, Quality, Novelty And Reproducibility:**

There are several issues with clarity as mentioned above, including description of the method, explanation of local isometry, and relationship to previous work. Due to the lack of clearly explained relationships between the proposed model and baseline methods, novelty is difficult to assess. Experimental results appear to be strong but the justification for each individual component of the method is weak. Reproducibility is also a concern as the model is complex yet code is not provided (the linked Github repository has only a placeholder readme as of the writing of this review).

**Strength And Weaknesses:**

Strengths
- The overall goal of learning disentangled representations from unsupervised data is important for interpretability and controllable generation.
- Strong experimental results on disentanglement metrics.

Weaknesses
- The proposed model is complex: it combines a VAE, a GAN, and a prototypical network. In order to justify each of these components, the experiments should isolate their relative contributions but I did not see such results in the paper.
- The paper is difficult to read. The description of the model is not sufficient to understand how all the components fit together. Specific  matters that remain unclear include: how the interventions were performed, how the support and query sets are generated, and how the traversal dimensions for the visualizations are selected.
- One of the stated key inductive biases is local isometry, but what exactly this means, why it is valuable, and its corresponding mathematical expressions require further elaboration.
- The relationship between the proposed ProtoVAE and previous VAE-style approaches to disentangled representation learning is not clearly explained. What aspects of the ProtoVAE are novel and what aspects are included in previous works?

**Summary Of The Paper:**

This paper proposes an approach, called ProtoVAE, that learns disentangled representations from unsupervised data. The overarching goal is to incorporate two inductive biases into the learned generative model: unique and consistent changes to the latent representations, and local isometry. ProtoVAE is based on a variational autoencoder, with several modifications. First, interventions are performed in the latent space such that both a reconstruction of the original image and a generated image corresponding to the intervened latent representation are produced. Second, a prototypical network is trained with pairs of images as input to predict both the intervention dimension and the magnitude of the change. Third, a GAN discriminator attempts to predict whether a latent representation was produced by the original inference network or if it was the result of an intervention. Experiments on dSprites, 3DShapes, MPI3D, and CelebA datasets show that the method is successfully able to learn disentangled representations.

**Summary Of The Review:**

Overall, the paper investigates an important goal, and there are potentially novel aspects contained in the proposed ProtoVAE model. However, the precise nature of the relationship between the ProtoVAE and previous baselines is still unclear in my mind. The disentanglement results are strong but the experimental justification for each of the components is also underdeveloped in the current version of the paper. I am hoping that these issues can be resolved during the discussion period.

---

> ### Author Response · Authors · 2022-11-17
> **Reply to Reviewer jevi**
>
> We thank the reviewer for their in-depth reviews.  We tried our best to make adjustments to the paper that addressed the points in the feedback.  While the adjustments are in the paper itself, we would like to point out the main replies we have to the feedback:
>
> *Component effectiveness*: The relative effectiveness and the contributions of each component is available in the ablation studies in the appendix. Our main contribution is the prototypical network and the need for the different components is justified in the ablation studies in Appendix C. Moveover, we have added some explanations for the need of the discriminator in the experimental section.
>
> *Difficulty of reading*: The paper and the method underwent significant writing to explain these details better (especially section 2)
>
> *Inductive bias*: The details on why Local Isometry is an effective bias and other details can be found in the [1] that was published before. We did not think that it was valuable to reiterate the details and would rather refer the reader to that paper for a detailed explanation. However, we have added an intuitive explanation of Local Isometry for the reader's convenience.
>
> *Novel contributions*: The novel part is the prototypical network and the corresponding constraints in the data space. The discriminator is novel in the sense that it allows for the model to perform interventions and is not solely used for disentanglement as in previous work. Otherwise (like we mention in the experimental section) the closest model to ours is the Factor VAE model with which we compare our results for all the datasets.
>
> We thank the reviewer again for taking the time to give this useful feedback and hope to hear back with additional feedback.
>
> [1] Horan, D., Richardson, E., & Weiss, Y. (2021). When Is Unsupervised Disentanglement Possible?. Advances in Neural Information Processing Systems, 34, 5150-5161.

---

### Decision · Program_Chairs · 2023-01-20

**Decision:**

Reject

**Justification For Why Not Higher Score:**

Major issues with clarity and presentation, that necessitated changing the paper significantly. Additionally, there are concerns regarding the significance of the results.

**Justification For Why Not Lower Score:**

N/A

**Metareview: Summary, Strengths And Weaknesses:**

This paper introduces an approach to unsupervised disentangling. While reviewers praised the positive experimental results on disentanglement metrics, the paper itself was unfortunately not in a state which left it ready for publication. All reviewers has issues with clarity and understanding, due to the presentation; in some cases this meant it was difficult for reviewers to be confident that the method itself was sound. While the authors have updated the paper to address these issues, as well as perceived issues with reproducibility, this represents a fairly major update to the paper and as such will likely require resubmission and further review.